# Deep Learning for Walking Behaviour Detection in Elderly People Using Smart Footwear

**DOI:** 10.3390/e23060777

**Published:** 2021-06-19

**Authors:** Rocío Aznar-Gimeno, Gorka Labata-Lezaun, Ana Adell-Lamora, David Abadía-Gallego, Rafael del-Hoyo-Alonso, Carlos González-Muñoz

**Affiliations:** Department of BigData and Cognitive Systems, Instituto Tecnológico de Aragón, ITAINNOVA, María de Luna 7-8, 50018 Zaragoza, Spain; glabata@itainnova.es (G.L.-L.); adell@itainnova.es (A.A.-L.); dabadia@itainnova.es (D.A.-G.); rdelhoyo@itainnova.es (R.d.-H.-A.); cgonzalez@itainnova.es (C.G.-M.)

**Keywords:** assistive technology, elderly people, wearable devices, smart footwear, deep learning, artificial neural networks

## Abstract

The increase in the proportion of elderly in Europe brings with it certain challenges that society needs to address, such as custodial care. We propose a scalable, easily modulated and live assistive technology system, based on a comfortable smart footwear capable of detecting walking behaviour, in order to prevent possible health problems in the elderly, facilitating their urban life as independently and safety as possible. This brings with it the challenge of handling the large amounts of data generated, transmitting and pre-processing that information and analysing it with the aim of obtaining useful information in real/near-real time. This is the basis of information theory. This work presents a complete system aiming at elderly people that can detect different user behaviours/events (sitting, standing without imbalance, standing with imbalance, walking, running, tripping) through information acquired from 20 types of sensor measurements (16 piezoelectric pressure sensors, one accelerometer returning reading for the 3 axis and one temperature sensor) and warn the relatives about possible risks in near-real time. For the detection of these events, a hierarchical structure of cascading binary models is designed and applied using artificial neural network (ANN) algorithms and deep learning techniques. The best models are achieved with convolutional layered ANN and multilayer perceptrons. The overall event detection performance achieves an average accuracy and area under the ROC curve of 0.84 and 0.96, respectively.

## 1. Introduction

### 1.1. Context

The proportion of elderly people in Europe has been increasing in recent years and is expected to follow a clear upward trend in the coming years, reaching 29.4% of the total population in 2050 [1]. This ageing population is due to falling fertility rates and increasing life expectancy, the latter due to numerous advances in science, technology, medicine and public health, combined with increased awareness of nutrition and personal hygiene [2,3]. Although the increase in demographic longevity can be seen as one of history’s great success stories, it has social consequences and challenges that need to be addressed, such as custodial care. One third of people over 75 have physical, mental or sensory impairments [4] and therefore need long-term custodial care. This care can be provided in institutional care or at home. Studies have shown that older people living in institutional care experience a higher level of dependency, loneliness and decreased life satisfaction and that they prefer to live in their own homes [5,6,7]. Living in their own home provides them with greater independence, reduces social isolation with a positive effect on the elderly [8]. However, ageing at home implies addressing certain aspects of home care.

The use of approaches and techniques for the care of the elderly has become an emerging challenge that needs to be addressed in a way that supports, facilitates and enables them to age with a better quality of life and as independently as possible. To this end, smart home systems [2,9,10,11,12] and assistive technologies (AT) [13,14] have been developed. They are based on the implementation of different sensors and devices (Internet of Things, IoT). However, their adoption may raise certain barriers [15] and concerns on the part of older adults related to how they perceive these technologies [16], such as privacy, ease of use, lack of training or suitability for everyday use [17]. This leads to high rates of dissatisfaction and abandonment of assistive technology and its use. Another aspect of abandonment is related to the design, aesthetics or unobtrusiveness of the device [18].

The integration of sensors and devices generates large amounts of real data and information, which brings with it the challenge of handling this data, pre-processing and analysing it in order to obtain useful information in real time. This is the basis of the fundamentals of information theory, which was conceived by Claude Shannon in 1948 [19]. Information theory is a subfield of mathematics that deals with the quantification of information, the representation of information and the ability of communication systems to transmit and process information. The need for this theoretical basis arose in the face of the increase in complexity and the massification of communication channels in the mid-20th century. Extrapolating it to the 21st century, with the development of concepts such as IoT, Artificial Intelligence, Big Data, Machine Learning, Deep Learning, the fundamentals of information theory remain basic foundations today. One of the important concepts of information theory is the quantification of the amount of information through the use of probabilities (“entropy”). This concept of information theory has had great contributions in areas such as machine learning and neural networks. In particular, the computation of information and entropy is a useful tool in machine learning and is used as a basis for techniques such as feature selection, decision tree construction, imbalance calculations in the target class distribution and, in general, when optimising classification models (e.g., artificial neural networks) considering cross-entropy as a loss function. The application of models based on artificial neural networks and particularly Deep Learning has become widespread in recent years due to its ability to automatically detect the most particular features of data, which has led to promising performance in many areas such as, in particular, sensor-based activity recognition [20,21,22,23] and in the application of smart homes and wearable devices [24,25,26].

Thanks to the fundamentals related to information theory, the miniaturisation of sensors and the improvement of data storage and transmission systems have been possible and is one of the reasons for the success of monitoring and pattern detection through IoT devices and sensors, particularly in the integration of fabrics and textiles (“smart fabrics/wearable”) [27,28,29,30]. Particularly, the data retrieved by sensors can be used to monitor the elderly in real time and predict their behaviour, preventing potential health problems, while providing them with independence and facilitating them urban living. Furthermore, ensuring that the electronics are fully integrated into the fabric ensures truly wearable products without discomfort.

The research problem of this article is based on the detection of walking behaviour of elderly people using wearable AT prototype for everyday use by using deep learning algorithms. This work is result of the European project MATUROLIFE (Metallisation of Textiles to make Urban living for Older people more Independent Fashionable) [31], which has been carried out within the framework of the Horizon 2020 (The EU-Horizon2020 (H2020-EU.2.1.3. Leadership in enabling and Leadership in enabling and industrial technologies—Advanced materials)). The project aim was to research, innovate and develop a more integrated assistive technology in textiles and fabrics through the use of advanced materials, allowing sensors and electronic devices to be fully integrated into intelligent fabrics in a discreet, fashionable and comfortable way [32,33,34,35,36]. The project studied the incorporation of sensors in three prototypes of AT for everyday use: clothing, furniture and footwear, that will make urban living for older adults easier, more independent, fashionable and comfortable. In the article we focus on the smart footwear prototype.

### 1.2. Related Work

Research and study of the walking behaviour detection through the use of footwear incorporating sensors (smart footwear) has been widely explored in last years [37]. Abnormal walking behaviour can indicate danger and detecting it can prevent potential health problems, such as injuries that can be caused by falls [38,39,40,41,42]. This is of great interest to the elderly population, as it allows them to lead a comfortable, more independent and safer urban life by monitoring their activity.

De Pinho et al. [43] presented the results of an experiment aimed at detecting 6 types of activities (walking straight, walking slope up, walking slope down, ascending stairs, descending stairs and sitting) from information retrieved from smart shoes. The experiment involved 11 participants, 2 of whom were elders. The classifier used a Random Forest algorithm with leave-one-out cross validation, achieving good average accuracy. A set of 12 features were considered as model inputs: 2 axis of the gyroscope, 2 axis of the magnetometer, 1 axis of the accelerometer, 4 force-sensitive resistors (FSRs), 2 Euler angles and the cumulative difference between samples of the barometer. el Achkar et al. [44] studied also the recognition of daily activities (level walking, sitting, standing, up/down stairs, up/down hill, elevator use) of older people. For this purpose, ten elderly people wearing the instrumented footwear system carried out the activities in a semi-structured protocol. The smart footwear included inertial and barometric pressure sensors, a sensorised insole to measure foot pressure and a box with electronics that strapped to the ankle. A decision tree incorporating rules inspired by movement biomechanics was applied as activity classification algorithm, achiving a high overall accuracy.

Zitouni et al. [45] designed a discreet, comfortable and highly effective device that is housed in the insole and a fall detection algorithm based on acceleration and time thresholds. Six subjects between 25 and 30 years of age were tested for possible falls that an elderly person may have while performing daily activities of daily living. They validated the proposed prototype and algorithm in real time (in a real public demonstration) confirming satisfactory performance. Montanini et al. [46] presented a low complexity and threshold-based methodology capable of detecting a fall and notifying a monitoring system. The smart shoes were equipped with 3 FSRs and a tri-axial accelerometer and tied to a belt an external processing unit box. These devices enabled the analysis of the subject’s motion and foot orientation, recognizing abnormal configurations. Laboratory tests involved 17 healthy subjects (aged between 21 and 55 years) and provided satisfactory performances in falls detection. The proposed method was also validated with two elderly users in a real-life scenario. Light et al. [47] mentioned the need for the use of monitoring systems for older people because of their high risk of falls and other mobility problems. They developed an optimized layout of pressure sensors for a smart- shoe fall monitoring application by analysing various machine learning algorithms with 10-fold cross validation that classify fall types. Subjects between the ages of 20 and 45 years participated in the data collection. The activities carried out in this experiment were falling-left, falling right, falling-forward, falling-backward, standing, walking, and kneeling down. Sim et al. [48] attached an accelerometer on the shoes (tongue) to detect fall in the elderly. 3 axis- acceleration signals were measured in three young subjects (2 young males, 1 young female, aged between 24 and 28). The fall types used in this study were the most common fall types in elderly people. The results of the fall detection algorithm showed that this shoe-based fall detection system had relatively high sensitivity.

### 1.3. Limitations of Existing Practices

The related studies have some limitations related to different aspects such as the comfort and usability of the smart footwear, the set of events capable of detection, the provision of a real-time detection and notification system, and the modularity and scalability of the system.

For greater comfort and ease of use of the footwear, it is desirable that sensors and electronic devices are fully integrated into the shoe without direct contact with the rest of the person’s body. This aspect is not fully addressed by some of the related works. The prototype of el Achkar et al. [44] housed an electronic box strapped directly to the ankle. Montanini et al. [46] did not fully integrate everything into the shoe either. Light et al. [47] mentioned that the insole was tied to the user’s leg using paper tapes, which could cause some discomfort, especially when removing these tapes from the subject’s leg. The design of the prototype of Sim et al. [48] also had some limitations. The accelerometer module of the prototype could easily detached because it was attached to the outside of shoes, as well as encumbering some activities, and the battery was slightly heavy. The authors suggested reducing the size of the module and embedding it under the insole, as well as incorporating piezoelectric elements to solve the problem with energy harvesting. All these factors represent a clear limitation, as they are less comfortable and intrusive devices that prevent certain activities from being carried out with complete normality.

Regarding the considered events, some of the studies focus on detecting more common events related to the user daily activity (sitting, walking, going up/down stairs, etc.) [43,44], while other studies focus only on immediate risk events such as falls [45,46,47,48]. However, there is a lack of studies proposing a system capable of detecting a broader spectrum of events, both hazard events and daily activity events.

Concerning the detection algorithms used, most studies proposed simple rule-based Machine Learning algorithms (decision tree [44], random forest [43]) or threshold-based methodologies [45,46]. Although in all cases the authors reported achieving good performance, they did not mention guarantees of modularity and scalability for the detection of new events or new functionalities and models. In addition, rule-based algorithms may be not very robust and have a higher risk of overfitting, with the risk of not generalising well to a different population.

There are also limitations in the related studies in terms of the provision of a system with practical real-time applicability. Most papers studied the scope of the system (algorithm and smart footwear) in terms of event detection but either did not validate it in real-time or did not provide a complete real-time detection and notification system. Montanini et al. [46] foreseed as future work the integration of a notification service for caregivers and Sim et al. [48] to develop a better smart-shoes system that shows the fall information on a smartphone and is therefore able to detect falls only with shoes and a smartphone.

### 1.4. Proposed Solution

We propose a system based on smart footwear capable of detecting different walking behaviours and warning the person responsible for the elderly person of possible risks in near-real time via a Telegram message. Figure 1 shows a general outline of the proposed system.

The system is able to detect 6 events (sitting, standing still without imbalance, standing with imbalance, walking, running and stumbling) from the information retrieved from the pressure, acceleration and temperature sensors incorporated in the smart footwear. For this purpose, Deep Learning techniques and neural networks algorithms are applied, following an architecture that allows the system to be easily modulated, scalable and robust. In addition, advanced materials were used in the design of the smart footwear, so that the sensors and electronic devices are fully integrated into the shoe in a discreet, elegant and comfortable way. This is important to encourage their use because, as mentioned above, suitability for everyday use, design, aesthetics, discretion and comfort are important aspects to their adoption. Many of the related studies introduced in the previous section do not fully address this aspect.

In conclusion, the system we propose includes a smart footwear, comfortable for daily use, that retrieves real-time information of the elderly person walking and is able to detect a wide set of events and warn the responsible person of possible dangers, allowing to act quickly to avoid potential health problems. Therefore, unlike most related works, in addition to studying the retrieved data and generating classification models based on deep learning, we developed a system useful for real practice that allows act and send notifications to the mobile phone in near real time. Furthermore, our system detects both events of possible immediate risk (imbalance, stumbling) and more common events in daily activity (walking, for example), making it a more complete behaviour detection system with greater practical interest. Another of the differential aspects that our proposal addresses is to ensure a scalable and easily modular system (it allows to be recreated with new functionalities) and alive, external to possible errors or failures in the sensors, for example.

To realise this whole system, the fundamentals of information theory were elementary and data analytics played a key role. Our work considered the methodological framework related to data analytics CRISP-DM (Cross-Industry Process for Data Mining) [49] which is based on an agile and iterative methodology whose approach consists of several interrelated phases: Business Understanding (understanding the context from the business perspective), Data Understanding (acquisition and exploration of the data), Data Preparation (pre-processing of the corresponding data for the subsequent use of models), Modelling (generation of Machine Learning and Artificial Intelligence models), Evaluation (evaluation of the results of the models related to the definition of the business objectives) and Deployment (deployment of the application). Specifically, the article focuses on the analysis and modelling of data from smart footwear sensors using Deep Learning techniques and artificial neural networks algorithms with cross-entropy loss function. The following sections present the proposed system in more detail, explaining the different modules it integrates, such as the data analysis module carried out, as well as the results and scopes obtained in the project.

## 2. Materials and Methods

### 2.1. System Architecture

The final objective of the designed prototype is to be able to use the information retrieved by the sensors implemented in the shoe in order to control the walking and movement behaviour of the user so that the relative responsible for the user or professional health carers can be alerted if a possible risk is detected. The system architecture achieving this is presented in Figure 2. The system architecture has been designed taking into account the potential scalability of the system with a possible growth potential. Kubernetes [50] is the platform used to manage the different components of the architecture.

The components involved in the architecture and therefore necessary to guarantee the information flow are: smart shoe, an Android smartphone for the user (in particular, having installed the MaturoApp application available on the PlayStore), mobile phone for the responsible person (contact person) with the Telegram application [51], internet connection (4G–5G) and an infrastructure (server) that supports the management, preprocessing and modelling of the data. The system is mainly composed of two information transmission processes: information retrieval flow from the user’s smart footwear to the database and information exploitation flow from the database to the end user (user relative).

The information retrieval process involves the following components: the smart footwear, the Android smartphone for the user (MaturoApp), a cloud data manager (MQTT [52]) and a database for storage. The process is as follows. The smart shoe generates information about the user gait through the measurements of the implemented sensors retrieved with an average frequency of 4 Hz. The raw data is coded by the PCB using scientific notation fixed-point coding. It is then transferred via Bluetooth to the user’s mobile phone (MaturoApp application), where the values of the pressure sensors can be displayed in real time. The mobile application transforms the received data chunks and transmits them via an MQTT message as follows:(1)$$\left\{“\mathrm{timestamp}”:“2020-10-25T17:12:24:6847”,“\mathrm{data}”:[X_{1},X_{2},X_{3},X_{4},X_{5},X_{6},X_{7},X_{8},X_{9},X_{10},X_{11},X_{12},X_{13},X_{14},X_{15},X_{16}]\right\}$$
where the timestamp and the values of the 16 pressure sensors ($X_{i}$) are displayed. These messages are grouped and inserted into an InfluxDB time series database [53].

Simultaneously, the information exploitation process is carried out. It involves the following components: the database with the retrieved information stored, the Data Analysis and Modelling module and the mobile phone of the contact person with the Telegram application. The process is as follows. The latest data stored in the database are retrieved with an appropriate frequency from the data Analysis module and preprocessed and prepared in a suitable way for the subsequent application of the artificial neural networks and Deep Learning models. The model output, as well as its timestamp, are stored in a table in the InfluxDB database. If the outcome indicates a risky event for the user, a Telegram message is sent to the emergency contact via the Telegram bot so an action can be taken. The information retrieval and exploitation system ensures near real-time event detection.

The following sections explain in detail the sensors implemented in the smart shoe, the experimental setup and data collection process necessary for the generation of the models and the data analysis module including sections on data pre-processing, model architecture and trained artificial neural networks.

### 2.2. Smart Footwear Design

Prototyped shoes were designed in 2 different models (male and female) and include sensorization and electronics in one shoe of each pair (right foot). In particular, the sensorized pairs of shoes that were made available for data collection and analysis were an European size 38 of the women’s shoe model and an European size 41 of the men’s shoe model.

The sensors implemented in the shoe retrieve information from three physical magnitudes: pressure (16 sensors), temperature (1) and acceleration (one measurement for each axis). The pressure sensors are piezoelectric sensors that are housed along the insole (designed with metallised textile) as presented in Figure 3. In particular, the material used for the produced insoles followed a coating process called electroless copper plating used for the selective metallization of textiles for electro-magnetic interference shielding [33]. The printed materials were used on a multilayer solution in which pressure sensors based on Printed Electronic technology are sandwiched between two layers with printed electrodes. The temperature sensor and the accelerometer are embedded in a printed circuit board (PCB). In addition to retrieving the temperature and acceleration measurements, the PCB is designed to connect and pre-process the data retrieved from the smart insole (pressure sensors), after a signal conditioning to convert pressure sensors signals into required values for the Analog Digital converters.

Besides the sensors, the shoes have a Bluetooth antenna, a battery and a micro-usb connector for charging. All these components and the PCB are housed in a 3D printed box incorporated in the heel area, for which it was necessary to drill a cavity. As it can be seen in Figure 4, the box is divided into two parts: one to hold the battery and the other to hold the PCB. The box has access to charging port and insole connector from outside.

Being the insole independent of the PCB allows the insoles to be easily removed and substituted with newer ones by simply unplugging them from the connector. Thus, users can easily replace the insoles if damaged or if newer versions arrive to market.

In order to check that no interactions with other signals occur that could affect the proper functioning of other surrounding devices, laboratory tests of the implemented sensorics were performed. The tests performed compared insoles produced with commercial textile sets (made with a flexible substrate circuit) and the ones produced with the metallized textiles with no variations detected regarding electro-magnetic interferences and resistance variations for the different forces applied. In addition, the communication system was designed and verified in such a way that it transmits information correctly and without error. Specifically, data transmission error rate is near null at the average distance a potential user could keep the mobile while walking (a pocket, on the hand, etc.), besides, the PCB is able to store several data chunks at an internal buffer and retransmit them on error till properly received by the mobile device.

### 2.3. Experimental Setup. Data Collection

#### 2.3.1. Events

In order to detect possible risks based on the user behaviour, a set of representative events of the gait to be modeled (supervised learning) was defined. The final classification models will allow the user behaviour to be related to one of the defined events. The events considered are the following:Sitting: sitting on a chair. May also include movements of the feet or the crossing of the legs.Standing still without imbalance: standing without moving forwards, backwards or sideways. May also include small foot taps.Standing with imbalance: includes lateral, frontal and random imbalances.Walking: includes different walking speeds, from slower to more normal.Running: running with a higher gait than walking.Stumbling: stumbling with the right foot. Includes both more violent and softer stumbles.

The selection criteria for the events was to consider a wide heterogeneous range of possible user behaviours including both more immediate hazard events and more common events of daily activity, in order to monitor and prevent possible health problems of the user. The stumbling and imbalance detection is important as these events indicate a possible risk of a fall and lack of body control by any user that may lead to a dangerous fall and negatively affect the user health. On the other hand, although the other events may relate to more common and less dangerous events in principle, their detection can also help to inform us of certain abnormal user behaviours in specific time periods. For example, the detection of the event “sitting” in a certain time period where the user usually walks may be an indication that the user has suffered a health problem (e.g., stroke), or the detection of the event “standing” without movement for a long period of time could indicate disorientation. As the data retrieval and exploitation system, discussed above, allows the detection of these events in a small time period (near real time), it enables emergency contacts to act rather quickly in order to avoid these potential health-related problems.

#### 2.3.2. Data Collection

For the application of the final classification models to detect the events presented above, it was necessary to generate and retrieve data (temporal information from the sensors) for each event. At the beginning the project activities were scheduled and addressed to a testing group of elderly participants at the village of Arnedo -La Rioja-. However, due to the COVID19 pandemic in 2020 that brought an initial lockdown across all Europe followed by mobility restrictions, they could only participate in the identification of needs and contribution of ideas to the product design and interaction teams. Therefore, as a consequence of these pandemic limitations, the generation or retrieval of (anonymised) data from the different defined events had to be finally performed by a group of participants from the project technical team. The group consisted of 3 people (2 women and 1 man), aged 26, 27 and 26 years, respectively, and weighting approximately 70 kg each, which remained stable throughout the study period. As they had different foot sizes, two of the subjects wore the same pair of shoes (male model) and one of the women wore the other pair (female model) throughout the study period.

The data collection process was as follows. The subject put the shoe on his right foot, logged into the MaturoApp application on his mobile phone and performed one of the events mentioned for a time. While this action lasted, the sensors implemented in the shoes were capturing acceleration, pressure and temperature values with the frequency mentioned above. This information was sent via Bluetooth to the Maturoapp application on the user’s mobile phone where the values of the pressure sensors could be visualised in real time. This data was stored in InfluxDB as described above with a manually defined tag identifying the subject, the event to which the data corresponded and the timestamp, in order to have complete traceability. This process was carried out multiple times by the 3 subjects and for the 6 events.

The database stored a total of 2.5 h of captured data. The first few minutes corresponded to initial recording tests in order to test and become familiar with the data capture system. Also, as explained later in the article, the models use as input the historical information for each time instant. Therefore, the first data captured in each recording were also not used in the generation of the model because they did not have sufficient historical information. Finally, the labelled dataset used for the generation of the models corresponded to a total of approximately 2 h of recording. The number of samples corresponding to each of the events is shown in Table 1.

Figure 5 shows an example of data generation for the event “walking”. The figure displays the person with the sensorized shoe and the image of the insole (shown in the MaturoApp) with the measurement information from the pressure sensors in real time. On the left of the figure is the person with the foot resting on the floor (active sensors with measurements in green) and on the right with the foot lifted (inactive sensors in red).

### 2.4. Data Analysis Module

#### Data Preprocessing

Once the data are captured and stored, a pre-processing of the data is carried out in order to unify the temporal information retrieved from the sensors before applying models.

Firstly, given that the acceleration sensors and the temperature sensor sent the data with a certain delay with respect to the pressure sensors, the times recorded by the accelerometer were assigned to the times of the pressure sensors. Thus all sensor readings ended up having the same time stamp each time a measurement was taken. Secondly, the maximum number of previous values that the model would use to predict the event was defined. After testing for computational speed and after having discussed and verified the time window to detect the event, a maximum of 32 previous values was chosen.

Therefore, the final data structure for each event was as follows (number of samples, 32, 20), the last component being the total number of sensor measurements. That is, each sample corresponded to a matrix of dimensions 32 × 20 of the form:(2)$$\left(\begin{array}{cccccccc} P_{t-1}^{0} & P_{t-1}^{1} & ... & P_{t-1}^{15} & T_{t-1}^{0} & A_{t-1}^{0} & ... & A_{t-1}^{2}\\ P_{t-2}^{0} & P_{t-2}^{1} & ... & P_{t-2}^{15} & T_{t-2}^{0} & A_{t-2}^{0} & ... & A_{t-2}^{2}\\ ... & ... & ... & ... & ... & ... & ... & ...\\ P_{t-32}^{0} & P_{t-32}^{1} & ... & P_{t-32}^{15} & T_{t-32}^{0} & A_{t-32}^{0} & ... & A_{t-32}^{2} \end{array}\right)$$
where *“P”* refers to the pressure sensor measurements, “*A*” to the acceleration and “*T*” to the temperature, the superscript corresponds, for each type, the sensor number and the subscript to the instant in time, being “*t*” the actual instant. Therefore, for each sample, historical information in the form of the Equation (Equation 2) was obtained.

Finally, the data associated with each of the events were divided into three separate data sets: training (60%), validation (20%) and test (20%), ensuring the same proportion of samples of each class (event) in each set. The training data was used for model training/tuning, the validation data was used for selection of the best model configuration (hyperparameter set) and the test data was used to provide unbiased evaluation metrics to give a generalized value of the performance of the chosen fitted model.

### 2.5. Model Architecture

For the prediction of the user state (defined events), different binary models were generated. The final outcome is the consequence of the application of these binary models in cascade following the hierarchical structure shown in Figure 6. The underlying idea was to start from more general binary models of behaviour that include more particular groups of events and, depending on their outcomes, to continue in the tree with more specific models, following a hierarchy.

A total of 5 types of binary models were generated: (1) binary model generated from recorded event information that determines whether the user remains seated or standing (standing still without imbalance, standing still with imbalance, stumbling, walking, running); (2) binary model generated from recorded standing event information that determines whether the user remains still (standing still without imbalance, standing still with imbalance) or moving (stumbling, walking, running); (3) binary model generated from the information of the recorded non-moving standing events that determines whether the user is unbalanced or not (standing still stable); (4) binary model generated from the information of the recorded moving events that determines whether the user stumbles or does not stumble (walking, running); (5) binary model generated from the information of the recorded non-stumbling moving events that determines whether the user is walking or running.

The training and validation of the different binary models is explained in the following section.

#### Artificial Neural Networks

Artificial neural networks architectures with different types of layers including dense layers, time-distributed dense layers, convolutional layers and long and short term memory (LSTM) layers were used to train the models.

The dense layer is the regular layer of the deep-connected neural network and the time-distributed dense layer applies the same dense layer to every temporal slice of an input. The structure of the convolutional layers has a connection between neurons that is not fully complete but parameters are shared between different neurons. This particular structure implies, on the one hand, the ability to learn general and invariant representations of the data and, on the other hand, the training of complex architectures with less computational time. The convolution structure used also allowed the use of pooling layers. The LSTM layers allow for a recurrence and learning of dependencies not only in the short term but also in the long term.

For the fitting of the models, the concept of “entropy” from information theory was used. In particular, cross-entropy was used as a loss function:(3)$$E=-\frac{1}{N}\sum_{i=1}^{N}y_{i}\cdot log\left(p\left(y_{i}\right)\right)+\left(1-y_{i}\right)\cdot log\left(1-p\left(y_{i}\right)\right)$$
where $y_{i}$ is the class (1 or 0), and $p(y_{i})$ the predicted probability of belonging class 1 for observation i out of N observations.

In order to avoid overfitting, regularisation techniques such as dropout and EarlyStopping were used so that the model would stop training if it did not improve within a certain number of iterations.

In the training of each binary event detection model, a hyperparameter search was performed, allowing to select the number of dense layers to be introduced, whether an LSTM layer or a convolutional layer was included as well as the value of the hyperparameters defining each of these neural network layers. The search space is presented in Table 2. The possibility to select or not the different sensors as inputs was also allowed. This implies a selection of features that may differ depending on the event to be modelled. Another parameter to choose was the number of previous timestamps for each event, this value being a maximum of 32 and a minimum of 4.

This hyperparameter search was carried out automatically using the framework called Optuna [54]. Optuna is a define-by-run API that allows users to construct the parameter search space dynamically and implements both searching and pruning strategies. Particularly, the Tree-structured Parzen Estimator (TPE) algorithm [55] was used. Thus, Optuna allowed training different models considering multiple combinations of hyperparameters. The selection of the best model configuration was carried out by the validation set. The metric considered was the area under the receiver operating characteristics curve (AUC). The discrimination ability of the final chosen models was calculated with the test set.

All data analysis and implementation of the models were performed using the Python programming language v. 3.8. [56].

## 3. Results

Table 3 shows the neural network architecture configuration of the best models for the 5 classification problems considered. Two of the models (stumble model and unbalanced model) selected a neural network with a convolutional layer and depth of 11 hidden layers as the best network configuration. For the stumbling problem 21 filters were chosen and 6 for the imbalance problem. For the rest of the problems, simpler architectures were chosen, namely multilayer perceptrons with one layer and 3 hidden layers (running vs. walking model). The model that discerns between running and walking and the one that discerns between standing with movement and standing still consider the 32 previous timestamps of each sensor as input; the model that discerns between sitting and standing and the unbalance model consider the previous 16 and the stumbling model the previous 4. Regarding the inputs of the best models, the Table shows the pressure sensors (*P*) and acceleration measurements (*A*) selected. Temperature was not selected in any of the models as an explanatory variable. In the case of the pressure sensors, the superscript indicates the sensor number, whose corresponding distribution along the insole is displayed in the Figure 3. The superscript in the acceleration indicates each of the three axes of the coordinate system. It is observed that the model that discerns between moving and still and the one that discerns between running and walking includes information from two acceleration axes as input to the model, while the rest of the models are fed with information from a single acceleration axis. Regarding the pressure sensors, in general the models considered as inputs a subset of pressure sensors housed along the entire insole. It could be noted that the model that discerns between standing and sitting considers more sensors from the bottom of the foot insole (heel) as inputs than the rest of the models. This result seems reasonable since the heel is the part of the foot that tends to bear a greater difference in load when standing compared to sitting.

This difference in the complexity of the neural networks between the problems addressed is sensible, as a stumble or an imbalance are more difficult to detect with the information provided by the wearable sensorised device than the other behaviours (sitting, standing without imbalance, walking, running), possibly due to their greater heterogeneity. The difference in the prior information needed may lie in the type of activity, some of them involving more continuous events over time (e.g., walking and running) and others shorter ones, such as stumbles.

The evaluation metrics of the best binary models are shown in Table 4. The metrics represented are accuracy, AUC, precision or positive predictive value (PPV), recall or sensitivity, f1-score, specificity and negative predictive value (NPV):(4)$$Accuracy=\frac{tp+tn}{tp+tn+fp+fn}$$
(5)$$Precision=\frac{tp}{tp+fp}$$
(6)$$Recall=\frac{tp}{tp+fn}$$
(7)$$Specificity=\frac{tn}{tn+fp}$$
(8)$$NPV=\frac{tn}{tn+fn}$$
(9)$$F1-score=2\cdot \frac{precision\cdot recall}{precision+recall}$$
where tp, tn, fp and fn are the number of true positives, true negatives, false positives and false negatives, respectively. The metric AUC is the area under ROC curve, being ROC curve a graphic representation of sensibility against 1-specificity depending on the discrimination threshold.

The model that discerns between sitting (class 0) and standing (class 1) and the one that discerns between standing without movement (class 0) and with movement (class 1) obtain a high performance close to 1. Although slightly lower, the model discerning between walking (class 0) or running (class 1) also achieves a high discriminative ability. However, the model discerning between no stumbling (class 0) and stumbling (class 1) and the one discerning between no unbalance (class 0) and unbalance (class 1) perform worse, with an AUC of 0.77 and 0.91 respectively. Consequently, the metrics show that problems dealing with stumble and imbalance detection are more difficult to model. As noted above, both problems were modeled with more complex network architectures.

In general, the metrics show that the binary models achieve a good discriminative ability. However, to calculate an evaluation metric for the general problem (detection of several events) it is necessary to apply the hierarchical structure (Figure 6) of the binary models. The average result achieved is an accuracy of 0.84 and an AUC of 0.96, which also indicates a remarkable overall performance.

## 4. Discussion and Conclusions

Although the increase in life expectancy could be considered one of history’s great success stories, it brings with it certain societal challenges that need to be addressed, such as the care of the elderly. Thanks to the advancement of technology (IoT, Big Data, Artificial Intelligence, etc.), highly innovative and attractive assistive technology (AT) products can be developed to enable a more independent, comfortable and safe life for the elderly.

The work presented is part of the result of the European project MATUROLIFE whose ultimate goal is to enable the elderly to age with the highest possible quality of life and independence through the implementation and development of an assistive technology integrated in wearable devices (clothing, furniture and footwear) in a discreet, fashionable and comfortable way. This sensorisation allows the remote monitoring of elderly people and the analysis of the large amount of data generated in order to prevent certain health problems. This article presents the prototype of the footwear (smart insole) that incorporates a total of 20 sensors that measure physical magnitudes such as temperature, pressure and acceleration. A scalable and easily modular system architecture was designed and implemented. Such architecture manages and updates the data retrieved from the smart shoes through an Android application (MaturoApp) via Bluetooth protocol, stores the information in a database on which the generated models are fed and sends a warning message via Telegram to the user’s contact person (responsible person) in the event of an indication of risk or anomalous behaviour. The fundamentals of information theory were essential to enable the system of consistent communication to transmit, process, analyse data and obtain useful information in near real time. The paper focuses especially on the part of detecting different walking behaviours by analysing and modelling the data retrieved from the smart footwear using deep learning techniques.

There are several studies that have explored algorithms that achieved good performance for human activity recognition based on smart footwear and focused on providing greater independence to elderly people. De Pinho Andre et al. [43] showed an average accuracy of 93.34%, el Achkar et al. [44] achieved a total algorithm precision of 97.41%, Montanini et al. [46] achieved an accuracy of 97.1%, Zitouni et al. [45] reached 100% sensibility and more than 93% sensitivity, Light et al. [47] achieved a 88% of accuracy approximately and Sim et al. [48] a 81.5% sensitivity. However, many of the studies focused on a single event such as falling [45,46,47,48] and others detected more common events related to the user daily activity but they do not include events of inmediate risk such as unbalancing, stumbling or falling [43,44].

Our proposal allows for the identification of 6 types of representative gait events that include events of immediate interest such as stumbling and imbalance and other more common events such as sitting, standing, walking or running. In the data collection process, some flexibility was allowed for in the conduct of these events. The criterion to consider these events was motivated in order to monitor and prevent possible user health problems by considering a wide range of possible user behaviours, both more immediate hazard events and more common everyday activity events that may also indicate abnormal behaviour depending on the patient and even the time of day. For example, although the event “walking” may be a completely normal event during the day, the detection of such an event at night may indicate abnormal behaviour of the patient at that time of the day when he should be sleeping. This may indicate disorientation and possible danger if prolonged over time.

Artificial neural network techniques and algorithms capable of detecting these events, which may be related to health problems such as disorientation, loss of control, among others, were explored and applied. In particular, 5 binary models were generated for the detection of such events through a hierarchical cascade structure. This cascade structure was designed so that the system could be easily modulated, allowing, for example, to be re-created with new binary models for the detection of more particular events, keeping the rest of the models or substituting only some of them. Optimisation in the training of the models allowed a choice of built-in sensors as inputs. The best models, which included different sensors as inputs, were stored sorted by performance. The aim of this was to ensure that the event detection system was always kept alive even in circumstances where a certain sensor stopped working, which may be possible in practice. Thus, if a sensor fails, the system uses the best model that does not use information from that sensor as input.

The results showed a high overall discrimination ability, reaching an average accuracy and AUC of 0.84 and 0.96, respectively. The worst performing binary models were those detecting stumble and imbalance with an AUC value of 0.77 and 0.91, respectively. This may be due to the fact that they are more difficult events to model than the others, as they are less constant and possibly more heterogeneous walking behaviours. This complexity is also reflected in the selected network architecture, where these models follow a network architecture with convolutional layers and considerable depth (11 hidden layers), while the rest of the models are multilayer perceptrons with 1 or 3 hidden layers.

Thanks to the fundamentals of information theory and the combination of different technologies such as sensing techniques, data acquisition and analytics, machine learning and deep learning, it is possible to improve the state of the art and develop new sensors and smarter systems. This is achieved by integrating intelligence techniques and deployment in wearables and related edge computing, where all related phases take place inside the sensorised device. This is the case of the work we present, which focuses on a wearable smart footwear comprising a scalable, easily modulated and live system that allows, through artificial neural network modelling, to detect with high accuracy a wide heterogeneity of walking behaviours and to warn the relatives or healthcare professionals about anomalous user behaviours so that they can act quickly. This system was designed in such a way that it can be implemented on any current embedded system with a lower CPU.

However, this work has some limitations and future work to consider. Due to the COVID19 pandemic in which we are immersed and the timelines set in the project, data collection by the end users (elderly population) was not possible. Consequently, data collection had to be carried out by the technical team with only 3 subjects of approximately the same age and weight. Other related studies included more heterogeneity in this regard and some involved elderly population in their experiments. De Pinho Andre et al. [43] used data from 11 subjects, two of whom were elders, el Alchkar et al. [44] involved ten elderly subjects (8 men, 2 women, age 65–75 years, weight 62–114 kg, height 162–184 cm), and Montanini et al. [46] conducted laboratory tests with 17 healthy subjects and demonstrated the effectiveness of their method with two older users in a real-life setting. Zitouni et al. [45] involved six subjects between 25 and 30 years of age, Light et al. [47] collected data from subjects aged between 20 and 45 years and Sim et al. [48] three young subjects (2 young males, 1 young female, aged between 24 and 28 years).

As future work, we propose to validate our system by including greater heterogeneity in the data, incorporating information from elderly population of different ages, weights and physical shape and in different environments where humidity, external temperature or the relief of the terrain may have an effect on the measurements. Thanks to the scalability and modularity that the designed system allows, another of the future lines of work to be explored could be to include clustering modules with the aim of grouping behaviours and applying specific models to each group. The application of modules for detecting changes in user behaviour (trend models) or the inclusion of more specific event models such as fall detection could also be studied.

## Figures and Tables

**Figure 1 entropy-23-00777-f001:**
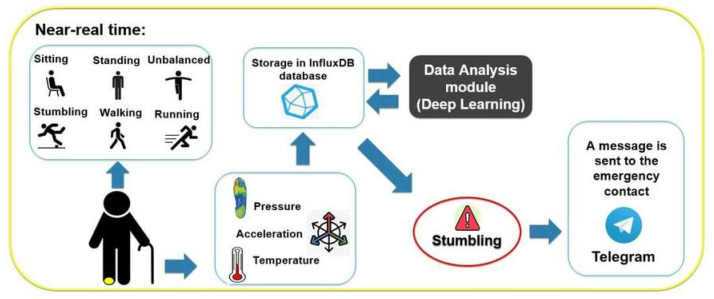
General outline of the proposed system.

**Figure 2 entropy-23-00777-f002:**
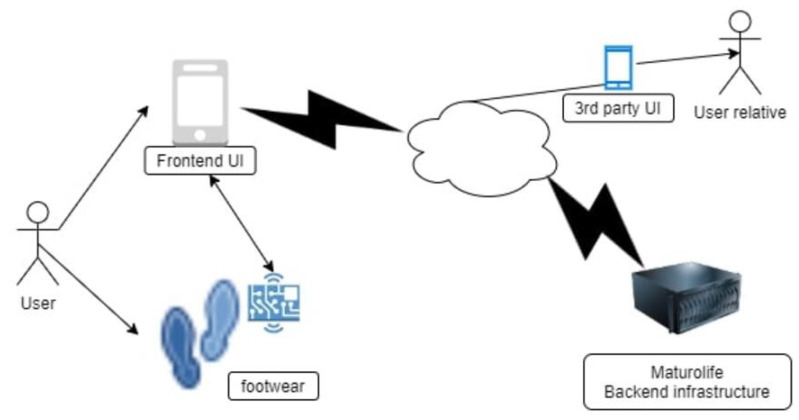
Scheme of system architecture.

**Figure 3 entropy-23-00777-f003:**
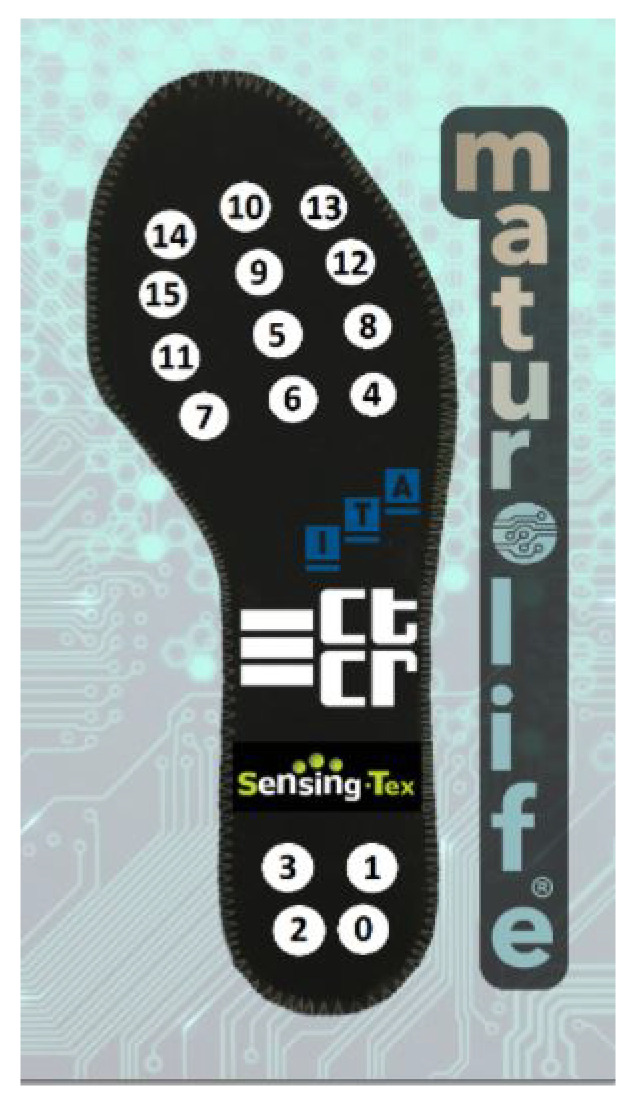
Location of the piezoelectric sensors on the insole.

**Figure 4 entropy-23-00777-f004:**
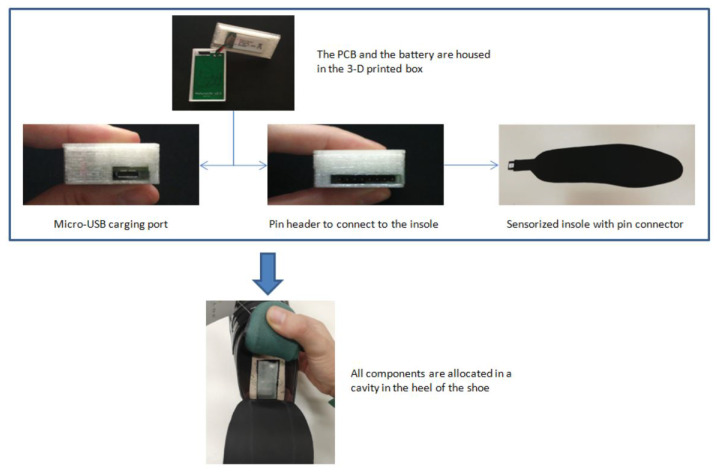
Components of the sensorised shoe.

**Figure 5 entropy-23-00777-f005:**
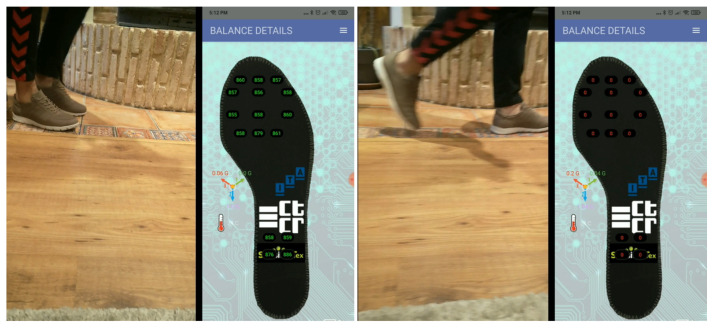
Recording data from sensorized shoe. **Left**: Shoe resting on the floor. **Right**: Shoe lifted.

**Figure 6 entropy-23-00777-f006:**
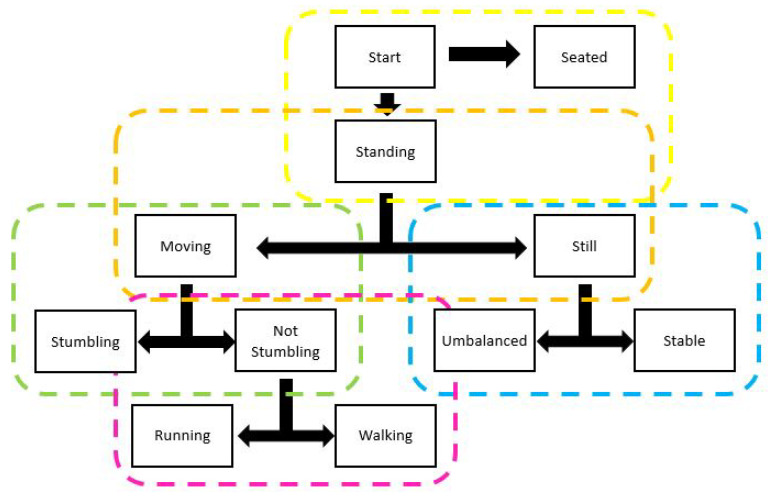
Hierarchical structure of models.

**Table 1 entropy-23-00777-t001:** Number of observations for each event.

Events	No of Observations
Sitting	3020
Standing still without imbalance	6920
Standing with imbalance	5230
Walking	9620
Running	3480
Stumbling	820

**Table 2 entropy-23-00777-t002:** Search space for each of the parameters when training the model.

Parameters	Search Space
No. of hidden Dense/Time Distr. Dense layers	[1, 11]
No. of units of each layer	[1, 64]
Activation function of each layer	{tanh,selu}
Use of Conv. layer	{True,False}
No. of filters in Conv.	[1, 50]
Size window in Conv.	[2, 32]
Activation function of Conv layer	{selu,sigmoid}
Use of pooling layers	[True,False]
Use of LSTM layer	[True,False]
No. of units of LSTM	[1, 50]
Learning rate	{0.1, 0.01, 0.001}
Optimizer	{sgd,adam,rmsprop}
Batch size	[1, 50]

**Table 3 entropy-23-00777-t003:** Best models configuration.

	Model Standing vs. Seated	Model Moving vs. Still	Model Stumbling vs. Not Stumbling	Model Unbalanced vs. Stable	Model Running vs. Walking
No. of hidden Dense/ Time Distr. Dense layers	1	1	11	11	3
No. of units of each layer	32	42	[26,54,16,38, 46,54,18,38, 36,54,1]	[32,4,38,50 34,38,22,6 24,58,58]	[46,6,9]
Activation function of each layer	selu	tanh	[tanh,selu,selu,selu, selu,tanh,selu,selu, selu,tanh,selu]	[tanh,tanh,selu,tanh, tanh,tanh,selu,tanh, selu,tanh,selu]	[selu,tanh,tanh]
Use of Conv. layer	False	False	True	True	False
No. of filters in Conv.	-	-	21	6	-
Size window in Conv.	-	-	10	4	-
Activation function of Conv layer	-	-	selu	sigmoid	-
Use of pooling layers	False	False	False	True	False
Use of LSTM layer	False	False	False	False	False
No. of units of LSTM	-	-	-	-	-
Learning rate	0.01	0.01	0.01	0.001	0.01
Optimizer	adam	sgd	sgd	adam	adam
Batch size	33	48	21	19	47
Previous timestamps	16	32	4	16	32
Selected sensors	{$P^{0},P^{1},P^{2},P^{5},P^{6},$	{$P^{3},P^{4},P^{6},P^{8},P^{10},$	{$P^{0},P^{1},P^{4},P^{7},P^{10},$	{$P^{0},P^{6},P^{9},P^{11},P^{12},$	{$P^{1},P^{3},P^{4},P^{6},P^{8},$
	$P^{7},P^{15},A^{0}$}	$P^{12},P^{13},A^{0},A^{2}$}	$P^{11},P^{12},P^{14},A^{1}$}	$P^{13},P^{15},A^{0}$}	$P^{11},P^{12},A^{0},A^{2}$}

**Table 4 entropy-23-00777-t004:** Metrics obtained by the best models.

	**Model Standing** ** vs. Seated**	**Model Moving** ** vs. Still**	**Model Stumbling** ** vs. Not Stumbling**	**Model Unbalanced** ** vs. Stable**	**Model Running** ** vs. Walking**
Accuracy	0.99	0.98	0.78	0.91	0.96
AUC	0.98	0.98	0.77	0.91	0.95
Precision	1	0.98	0.64	0.91	0.95
Recall	1	0.99	0.75	0.89	0.91
F1-score	1	0.98	0.69	0.9	0.93
Specificity	0.97	0.98	0.79	0.93	0.98
NPV	0.97	0.99	0.86	0.91	0.97

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
