# Peer review of "Deep Learning for Walking Behaviour Detection in Elderly People Using Smart Footwear"

_entropy, 2021, doi:10.3390/e23060777_

Round 1
Reviewer 1 Report
The paper is well written, it suffers from lack of true experimental results (as admitted by the authors themselves in the conclusions), but I believe it can be published, being the result of a European project. The work itself was very complete (from hardware to software).
In sections 1 and 4 the authors stress the importance of information theory in their work, but the contribution seems to be confined to the choice of the loss function, i.e. cross-entropy, which is fairly common and does not need explanations. Some sentences on this topic can be completely removed to shorten the paper.
Some details should be however clarified.
The role of temperature as a sensor is one example: at line 323 the authors write that the optimization allowed to select which sensors to use to detect a given event, but then they say nothing else, and I wonder how temperature could be relevant to detect, for example, stumbling. Can the authors list in table 2 which sensors were actually selected by the optimizer for each event?
At line 406 the authors write that they can find abnormalities depending on the time of the day, but it is not clear how this is possible, since in eqn (2) there is no time information in the matrix.
The authors write that they used data collected using three "patients" and in line 246 specify that "Total data capture time was approximately 2.5 hours". Do they mean 2.5 hours for each "patient" (7.5 hours) or really 2.5 hours in total? If this second case is correct, then the matrix they work with should contain 45000 rows/points, this should be clearly specified. The number of events to be detected is relatively high (11) and I wonder how many realizations of each event were present in the dataset. In Table 3 accuracy for example is given equal to 0.99 for classifying standing versus seating, but this makes sense only if at least 1000 events are present in the dataset, possibly 500 standing and 500 seating. The authors could give more details on the description of the dataset and the number of each simulated/reproduced events by each of the "patients".
Typos/sentences to change:
Line 155: "material implemented" -> "material used/produced/chosen"
Line 169: "independent to " -> "independent of"
Figure 3: top right "3th" -> "3rd"
Line 198: "bluetooth" -> "Bluetooth"
Line 258: "and on the right without rest it " to be rephrased, not clear
Line 280: "Arquitecture" -> "Architecture"
Tables 1 and 2: "hiden" -> "hidden", "F.act." -> "Act.Fcn" (or similar, for activation function)
Reviewer 2 Report
- Introduction has provided a detailed background on the sue of smart sensors in various disciplines. However, the research problem of this article is related to the walking behaviour detection in elderly people using smart footwear. Therefore, it is important to first discuss the related work on this particular problem. A dedicated subsection can be used for this purpose (sub-section 1.1). Based on the discussion of relate work, the limitations of existing practices must be highlighted (subsection 1.2). The contributions of the article should be explicitly stated in Section 1.3. It should also be discussed that how the proposed solution is going to address the limitations in Section 1.2. Moreover, the major steps of the proposed system should be illustrated through a diagram and appropriately explained in the text. Finally, there should be a discussion at the end of introduction that how the proposed system is beneficial in terms of various performance factors.
- Section 2 provides seven subsections. It is suggested to re-organize the presentation. A top down approach is always better in such scenarios. It implies that first the system level presentation should be provided. The system level presentation should provide the structure of proposed system in terms of various modules. What are different modules and how they are connected with each other ?. Once the structural discussion is finished, it is required to describe the behavior of each module. In the current form, the presentation is poor and it is very hard to understand the proposed system in a systematic way.
- Section 3 is weak in terms of contribution and presentation. For example: (a) experimental setup is not discussed (b) motivation and description of case studies is missing (c) An explicit description of various performance attributes is not provided. (d) A performance comparison with stat-of-the-art in terms of various performance attributes is not provided.
In addition to aforementioned major concerns, the authors are suggested to review the entire paper from English language presentation point of view. For example, I am just highlighting few issues in abstract. Authors are suggested to thoroughly analyze all the section in a similar way.
- The significance of proposal is not provided in abstract.
- The abbreviation “AUC” is used in abstract without definition.
- Line 9 of the abstract says that “from 20 measurement sensors (16 piezoelectric pressure sensors, one accelerometer and one temperature sensor)……”. In parenthesis, the description of only 16 sensors is provided.
- Line 11 of the abstract says that “For the detection of these events, a hierarchical structure of cascading binary models (5). Here the number 5 is not providing any sense.
- The use of tenses in abstract is mixed (both present and past tenses are used). The uniformity is generally considered a better choice.
Round 2
Reviewer 2 Report
All the raised concerned have been addressed.
The article can be published in its current form.
This manuscript is a resubmission of an earlier submission. The following is a list of the peer review reports and author responses from that submission.